# Chemical Composition of Essential Oil from *Citrus reticulata* Blanco cv. Chachiensis (Chachi) and Its Anti-Mosquito Activity against Pyrethroid-Resistant *Aedes albopictus*

**DOI:** 10.3390/insects15050345

**Published:** 2024-05-11

**Authors:** Jifan Cao, Wende Zheng, Baizhong Chen, Zhenping Yan, Xiaowen Tang, Jiahao Li, Zhen Zhang, Song Ang, Chen Li, Rihui Wu, Panpan Wu, Wen-Hua Chen

**Affiliations:** 1School of Pharmacy and Food Engineering, Wuyi University, Jiangmen 529020, China; caojifan2020@163.com (J.C.); 15875045599@163.com (W.Z.); yzp1297209652@163.com (Z.Y.); wyutxw@126.com (X.T.); 18676125540@163.com (J.L.); z1833918@163.com (Z.Z.); jnuangsong@126.com (S.A.); wyuchemlc@126.com (C.L.); wyuchemwrh@126.com (R.W.); 2International Healthcare Innovation Institute (Jiangmen), Jiangmen 529040, China; 3Guangdong Xinbaotang Biotechnology Co., Ltd., Jiangmen 529100, China; cbz1339@126.com

**Keywords:** *Citrus reticulata* Blanco cv. Chachiensis (Chachi), essential oil, larvicidal activity, enzymatic activity, internal microbiota

## Abstract

**Simple Summary:**

The essential oil (EO) of *Citrus reticulata* Blanco cv. Chachiensis (Chachi) exhibited significant larvicidal activity. The EO from semi-mature fruits and its main components were able to control mosquitoes by effectively inhibiting acetylcholinesterase and three detoxification enzymes and significantly altering the diversity of the internal microbiota in mosquitoes.

**Abstract:**

The overuse of synthetic insecticides has led to various negative consequences, including insecticide resistance, environmental pollution, and harm to public health. This may be ameliorated by using insecticides derived from botanical sources. The primary objective of this study was to evaluate the anti-mosquito activity of the essential oil (EO) of *Citrus reticulata* Blanco cv. Chachiensis (Chachi) (referred to as CRB) at immature, semi-mature, and mature stages. The chemical compositions of the CRB EO were analyzed using GC-MS. The main components were identified to be D-limonene and *γ*-terpinene. The contents of D-limonene at the immature, semi-mature, and mature stages were 62.35%, 76.72%, and 73.15%, respectively; the corresponding contents of *γ*-terpinene were 14.26%, 11.04%, and 11.27%, respectively. In addition, the corresponding contents of a characteristic component, methyl 2-aminobenzoate, were 4.95%, 1.93%, and 2.15%, respectively. CRB EO exhibited significant larvicidal activity against *Aedes albopictus* (*Ae. albopictus*, Diptera: Culicidae), with the 50% lethal doses being 65.32, 61.47, and 65.91 mg/L for immature, semi-mature, and mature CRB EO, respectively. CRB EO was able to inhibit acetylcholinesterase and three detoxification enzymes, significantly reduce the diversity of internal microbiota in mosquitoes, and decrease the relative abundance of core species within the microbiota. The present results may provide novel insights into the utilization of plant-derived essential oils in anti-mosquitoes.

## 1. Introduction

Mosquito-borne viral diseases have persistently posed a significant health challenge and exhibited increasing prevalence in both tropical and temperate regions [1]. *Aedes albopictus* (Diptera: Culicidae) is the primary vector for numerous disease-causing arboviruses, including yellow fever virus [2], dengue fever virus [2], chikungunya virus [3], and Zika virus [4]. Currently, the primary method of managing mosquito vectors is to use chemical insecticides, such as pyrethroid, organophosphate, and organochlorine compounds [5]. However, the constant and extensive use of chemical insecticides has led to the development of mosquito resistance [6,7] and posed potential toxic effects on public health and environments [8]. Meanwhile, studies have documented the resistance of mosquitoes to insecticides, which may continue to pose challenges for communities in coming decades [9]. Thus, it is imperative to develop innovative agents for the control of mosquito vectors [7]. Essential oil (EO) has garnered significant interest due to its potential biological effects, including antibacterial, antiviral, and insecticidal activity [10,11,12]. Furthermore, studies have demonstrated that plant-derived bioactive metabolites in EO may act as pesticide synergists to increase insecticidal efficacy [13].

*Citrus reticulata* Blanco cv. Chachiensis (Chachi) (referred to as CRB) is a member of the Rutaceae family, and it is recognized as one of the most important commercial tangerines in Guangdong Province, China [14]. The dried peels of *C. reticulata* and its cultivars, that is, Pericarpium Citri Reticulatae (PCR, Guang Chen Pi in Chinese), are highly valued in China as a precious traditional Chinese medicine, food, and spice ingredients [14,15]. Pharmacological studies have demonstrated the significant biological potentials of *C. reticulata*, including antioxidant, antitumor, antimicrobial, antiatherosclerosis, and anti-inflammatory activity [16,17,18]. However, the anti-mosquito effect of the constituents in *C. reticulata* peel EO has been rarely investigated [19]. In a previous study [20], we demonstrated the potent insecticidal activity of EO derived from PCR against *Aedes albopictus* larvae. CRB is considered the primary source of genuine PCR [21]. Compared with PCR, CRB has the advantages of low cost and ready availability. Furthermore, it has been reported that fruits that are harvested at different stages of ripening possess considerably varied chemical compositions and bioactive profiles [22,23,24].

In the work reported herein, we sought to prepare EO by hydrodistilling the peels of CRB at immature, semi-mature, and mature stages (Figure 1), evaluate its anti-mosquito activity, and explore the probable mechanism of biological action. The outcome of this study is expected to provide support for the prevention and management of drug-resistant mosquitoes using CRB EO.

## 2. Materials and Methods

### 2.1. Plant Material and Chemicals

The CRB used in this study was provided by Guangdong Xinbaotang Biotechnology Co. Ltd. (Xinhui, Jiangmen, Guangdong, China) and identified by Prof. Junxia Zheng at the Guangdong University of Technology. The immature, semi-mature, and mature peels were harvested in September, October and December, respectively. Commercially purchased reagents included a custom standard solution of C7-C40 n-alkanes (reagent brand: o2si, analytical reagents) from ANPEL Laboratory Technologies (Shanghai, China), *γ*-terpinene (purity: 95%) and *p*-cymene (purity: 98%) from Meryer Chemical Technology Co., Ltd. (Shanghai, China), D-limonene (purity: 95%) from Tokyo Chemical Industry Co., Ltd. (Shanghai, China), methyl 2-(methylamino)benzoate (purity: 98%) from Shanghai Aladdin Bio-Chem Technology Co., Ltd. (Shanghai, China), *α*-terpineol (purity: 98%) from Saen Chemical Technology (Shanghai) Co., Ltd. (Shanghai, China), and deltamethrin from J&K Scientific (Beijing, China). All the other reagents were purchased from Tansoole (Shanghai, China).

### 2.2. Preparation of CRB EO

The peels of CRB were hydrodistilled at 100 °C at the ratio of 4:1 water to plant (*w*/*w*). The extracted EO was then dried over anhydrous sodium sulfate, filtered, weighed, and stored in dark sealed vials at 4 °C. Three types of EO were obtained from immature peel, semi-mature peel, and mature peel.

### 2.3. Mosquitoes

The pyrethroid-resistant strain was sourced from the Jiangmen Center for Disease Control and Prevention, Guangdong Province, China, and maintained consecutively at the International Healthcare Innovation Institute, Jiangmen, China. The mosquitoes were reared under the conditions of a 14:10 light/dark photoperiod and 70 ± 5% relative humidity at 26 ± 2 °C. Larvae and adults were fed daily with fish food and a 5% glucose solution, respectively. Fourth instar larvae and 2- to 5-day-old female mosquitoes were utilized in this study [25,26]. After 7- to 10-day-old female adult mosquitoes mated with male adult mosquitoes, they were fed sterile defibrillated sheep blood (purchased from Guangzhou Xiangbo Biotechnology Co., Ltd., Guangzhou, China.) using an artificial homemade feeding device.

### 2.4. Chemical Analysis of CRB EO

Gas chromatography-mass spectrometry (GC-MS) analysis was carried out on a Thermo Scientific TRACE 1300 Gas Chromatograph linked to an ISQ Qd mass spectrometer and equipped with a TG-5 MS capillary column (30 m × 0.25 mm i. d., 0.25 μm film thickness, Thermo Scientific, Shanghai, China). Helium was used as a carrier gas at the flow rate of 1 mL/min. Initially, the temperature was set at 35 °C and maintained for 5 min. Subsequently, it was ramped up to 120 °C at a rate of 3 °C/min and held for 3 min. Finally, the temperature was increased to 280 °C at a rate of 10 °C/min and held for 20 min. The injector and detector were kept at 250 °C and 280 °C, respectively. Furthermore, the temperature of the ion source was set at 320 °C. The mass spectra were recorded in an electron impact ionization (EI) mode at 70 eV, with a scan range spanning from 35 to 500 *m*/*z* [7,26].

The components of CRB EO were identified by comparing the search results of the NIST mass spectral library with commercially available standards and Kovats retention indices (RI). The RI values were determined relative to the retention time of a C8–C40 n-alkane standard and calculated according to Formula (1). Then, the calculated RI values were compared with the RI values provided in the NIST Chemistry WebBook [26].
(1)$$\mathrm{RI}=100n+\frac{100(t_{x}-t_{n})}{t_{n+1}-t_{n}}$$
where in t_x_, t_n_, and t_n+1_ denote the retention time of the analyzed component, n-alkane having n carbon number, and n-alkane having n + 1 carbon number, respectively; t_n_ < t_x_ < t_n+1_.

### 2.5. Measurement of Insecticide Susceptibility

An immersion method [25,27] and a CDC bottle bioassay [28] were performed to evaluate the toxicity of deltamethrin on mosquitoes. In short, deltamethrin was dissolved in acetone at concentrations that caused 10% and 90% mortality. Twenty larvae or non-blood-fed female mosquitoes were exposed to pyrethroids. Mortality was determined after 24 h of exposure, and the LC_50_ was calculated using an SPSS PROBIT model. Here, LC_50_ represents the lethal concentration of deltamethrin that kills 50% of mosquitoes.
(2)$$Resistance\;Ratio\;(RR)=\frac{{LC}_{50}(Pyr-R)}{{LC}_{50}(Lab-S)}$$

RR ≤ 3 is suggested for a susceptible resistance level, 3 < RR ≤ 10 is suggested for low-resistance, 10 < RR ≤ 20 is suggested for mid-resistance, and RR > 20 is suggested for high-resistance.

### 2.6. Larvicidal Assay

The larvicidal activity was assessed using an immersion method, according to the standard procedures recommended by the World Health Organization, with slight modifications [25,27]. Firstly, a stock solution of CRB EO at a concentration of 10 mg/mL in acetone was prepared. Secondly, the stock solution was diluted to 1–10 g/L. Thirdly, 1 mL of the diluted solution was added to 99 mL of 1% acetone aqueous solution (*v*/*v*), and the final test concentration was 10–100 mg/L. For the preliminary screening, 20 larvae were transferred into the testing solution without feeding during the test period. Mortality was determined after 24 h of exposure, and the lethal doses were determined from three replicates for each sample at each concentration. Larval mortality for each concentration was recorded after 24 h of continuous exposure to treatments. LC_50_ and LC_90_ were calculated using an SPSS PROBIT model. Here, LC_90_ represents the lethal concentration of CRB EO that kills 90% of mosquitoes. A blank reference without the treatment was used as a control. The adjusted mortality was calculated according to Formula (3) [25].
(3)$$\mathrm{Adjusted}\;\mathrm{mortality}\;\left(\%\right)=\frac{\mathrm{mortality}\;\mathrm{of}\;\mathrm{testing}\;\mathrm{group}-\mathrm{mortality}\;\mathrm{of}\;\mathrm{blank}\;\mathrm{group}}{1-\mathrm{mortality}\;\mathrm{of}\;\mathrm{blank}\;\mathrm{group}}\times 100\%$$

### 2.7. Mosquito-Adulticidal Activity

The toxicity of CRB EO against pyrethroid-resistant *Ae. albopictus* strains was evaluated using a bottle bioassay from the Centers for Disease Control and Prevention (CDC) [28]. Firstly, a stock solution of CRB EO and its main components was prepared in acetone, and a series of concentrations was prepared by appropriately diluting this stock solution. Secondly, 1 mL of the test solution was transferred into a 250 mL Wheaton bottle. While shaking and rotating the bottle, the solvent was evaporated at room temperature, and a uniform film was formed on the inner surface of the bottle. Thirdly, twenty non-blood-fed female mosquitoes (2–5 days old) were introduced into each bottle for 2 h. Then, the mosquitoes were transferred to culture cups and cultured in the incubator. After 24 h, the mortality checks were assessed. Acetone was used as a negative control.

### 2.8. Enzymatic Activity

The activity of CRB EO toward acetylcholinesterase (AChE) was assessed according to the procedures reported by Ellman et al. [29], employing acetylthiocholine iodide (ASChI) as a substrate and dithiobisnitrobenzoic acid (DTNB) as a chromogen. The activity of CRB EO toward glutathione-S-transferase (GST) was determined using the methods described by Polson et al. [30] and 1-chloro-2,4-dinitrobenzene (CDNB) as a substrate. The activity of CRB EO toward mixed function oxidase (MFO) was measured using tetramethyl-benzidine dihydrochloride (TMBZ) and H_2_O_2_. The esterase activity was measured according to the method outlined by Azratul-Hizayu et al. [31]. Twenty pyrethroid-resistant adults were homogenized in 1 mL of phosphate buffer (0.1 M, pH 7.8). Insect homogenate was centrifuged at 17,000× *g* and 4 °C for 15 min, and the resulting supernatant was transferred to a clean centrifuge tube for use as an enzyme source. Protein levels were determined using the Bradford Protein Assay Kit. The enzymatic assays were carried out in triplicate using a clear 96-well flat-bottom microplate.

### 2.9. Internal Microbiota Community

Using a CDC bottle bioassay, adult pyrethroid-resistant Ae. albopictus were exposed to semi-mature *EO*, p-cymene, D-limonene, and γ-terpinene, respectively, with acetone exposure as a control. After the pretreatment with *CRB EO* was completed, mosquitoes were frozen at −20 °C. Then, mosquitoes were sterilized by rinsing them with ethanol three times under sterile conditions and then frozen at −80 °C until metagenomic analyses were performed. The metagenomic and data analyses were carried out at Biomarker Technologies Co., Ltd. (Beijing, China).

#### 2.9.1. DNA Extraction

Each sample containing six mosquitoes was subjected to extra-genomic extraction using a TGuide S96 Magnetic Soil/Instrumental DNA Kit (Tiangen Biotech (Beijing) Co., Ltd., Beijing, China), according to manufacturer instructions. The Qubit dsDNA HS Assay Kit and a Qubit 4.0 Fluorometer (Invitrogen, Thermo Fisher Scientific, Waltham, MA, USA) were used to measure the DNA concentration. The integrity was evaluated through 1.8% agarose gel electrophoresis.

#### 2.9.2. Amplicon Sequencing

The hypervariable regions V3-V4 of the 16S rRNA gene were amplified by using primers 338F (ACTCCTACGGGGAGGCAGCA) and 806R (GGACTACHVGGGTWTCTAAT) extracted from each sample. The forward and reverse 16S primers were tailed with sample-specific Illumina index sequences for deep sequencing. Following the individual quantification steps, amplicons were pooled in equal amounts. To construct libraries, sequencing was performed using an Illumina novaseq 6000 (Illumina, San Diego, CA, USA).

#### 2.9.3. Bioinformatic Analysis

According to the quality of the single nucleotide, raw data were primarily filtered using Trimmomatic [32] (version 0.33). The identification and removal of primer sequences were processed by Cutadapt [33] (version 1.9.1). PE reads obtained from previous steps were assembled using USEARCH (version 10) and followed by chimera removal using UCHIME [32] (version 8.1). The high-quality reads generated from the above steps were used in the subsequent analysis. Sequences with similarities of ≥97% were clustered into the same operational taxonomic unit (OTU) using USEARCH (v10.0), and the OTUs with re-abundances of <0.005% were filtered. Taxonomy annotation of the OTUs was performed based on the Naive Bayes classifier in QIIME2 using the SILVA database (release 132) with a confidence threshold of 70%.

### 2.10. Statistical Analysis

The LC_50_ and LC_90_ values of Ae. albopictus with 95% confidence intervals were calculated using a Probit analysis of SPSS 25.0 software. Significant differences between treatment groups were analyzed using one-way ANOVA and Tukey’s post hoc test using Graphpad Prism 8.0 (San Diego, CA, USA).

## 3. Results and Discussion

### 3.1. Composition of CRB EO

The chemical compositions of CRB EO were identified using GC-MS. Table 1 summarized the components, retention time, retention indices, and percentages at different ripening stages. As a result, twenty-eight common components were identified in the CRB EO. Unexpectedly, CRB EO had fewer chemical components than 5-year PCR EO. Some low-content compounds in 5-year PCR EO were not detected in CRB EO, whereas the high-content components are similar [20]. D-Limonene emerged as a predominant secondary metabolite, with contents ranging from 62.35 to 76.72%, and the highest content was observed in the semi-mature peel. Furthermore, the EO extracted from the peel at the three maturation stages contained high contents of *p*-cymene (1.39–5.35%), *γ*-terpinene (11.04–14.26%), methyl 2-aminobenzoate (1.93–4.95%), and *a*-terpineol (0.46–2.42%) (Figure 2). The heat map in Figure 3 shows the relative abundance levels of the components in the CRB EO. Here, an increase in the content is indicated by transitioning from purple to red.

### 3.2. Insecticide Susceptibility

To quantify the resistance of each strain to insecticides, we measured the LC_50_ values of deltamethrin for both laboratory-susceptible strains (Lab-S) and pyrethroid-resistant strains (Pyr-S) (Table 2). Both larvae and adults displayed resistance to deltamethrin. The resistance ratio (RR) of permethrin was approximately 13.51 for adults and 7.73 for larvae, indicating a greater resistance in adults than in larvae.

### 3.3. Larvicidal Activity of CRB EO and Its Main Chemical Components

Developing highly effective insecticides from botanicals is an interesting route to tackle the fast-growing resistance of mosquitoes to synthetic compounds [34,35]. It is reported that EO extracted from other CRB exhibits high activity against mosquitoes [20]. In this study, CRB EO exhibited considerable effectiveness (LC_50_ = 61.47–65.91 mg/L and LC_90_ = 77.40–100.21 mg/L) in controlling fourth-instar pyrethroid-resistant larvae, surpassing the efficacy of ethanol extract from Citrus sinensis orange peel (LC_50_ = 264.26 mg/L) [36] (Table 3). In previous studies, PCR EO was toxic to *Ae. Albopictus* laboratory susceptive larvae, with LC_50_ values ranging from 54.50 mg/L to 68.23 mg/L [20]. CRB EO exhibits comparable larvicidal activity against *Ae. albopictus* larvae with PCR EO, indicating its potential in the control of mosquitoes. Although the World Health Organization does not specify the effective concentration for larvicidal activity, one component with an LC_50_ of less than 100 mg/L is generally considered active [37].

Based on the LC_50_, the toxicity of the main components in CRB EO toward mosquito larvae decreases in the order of D-limonene > *p*-cymene > *γ*-terpinene > methyl 2-(methyl-amino)benzoate > *α*-terpineol. In addition, D-limonene (LC_50_ = 64.33 mg/L), *p*-cymene (LC_50_ = 72.31 mg/L), and *γ*-terpinene (LC_50_ = 75.78 mg/L) exhibited higher activity than the other tested components. The results suggest that those chemicals are highly lethal to *Ae. albopictus* larvae, and it is likely that they are responsible for the larvicidal activity observed in the CRB EO. The larvicidal activity of D-limonene in pyrethroid-resistant stain (LC_50_ = 64.33 mg/L) is comparable to that in pyrethroid-susceptive stain (LC_50_ = 56.17 mg/L) of *Ae. Albopictus* [20].

### 3.4. Adulticidal Activity of CRB EO

The toxicity of CRB EO against Pyr-R strains of *Ae. albopictus* adults was summarized in Table 4. The results revealed that CRB EO showed low toxicity against adult mosquitoes. Previous studies have reported the toxicity of many plant EO against mosquito larvae [38]. However, the efficiency of EO derived from *C. reticulata* against adult mosquitoes has been rarely reported. To develop the application of CRB EO, it is necessary to explore its potential in adult mosquito control and its use as a synergist.

### 3.5. Enzymatic Activity

As a serine esterase (hydrolase) that is predominantly found in nerve synapses, AChE is a crucial enzyme responsible for terminating nerve impulses by catalyzing the hydrolysis of acetylcholine, a neurotransmitter in nervous systems [39]. AChE has been identified as a crucial target for the larvicidal activity of plant essential oil insecticides [19,40]. The main detoxification enzymes in mosquitoes include carboxylesterase (EST), glutathione-S-transferase (GST), and cytochrome P450 enzymes. Cytochrome P450 enzymes serve as the terminal oxidase of multifunctional oxidase (MFO), and thus, the activity of MFO indirectly reflects the activity of cytochrome P450 enzymes [41,42].

The inhibitory activity of CRB EO toward the four enzymes is shown in Figure 4. It can be seen that CRB EO exhibited varying inhibition on the four enzymes. The semi-mature EO and its components were able to inhibit the activity of AChE (inhibition 27.46%, F_5,12_ = 188.1, *p* < 0.0001, Appendix A), EST (inhibition 31.81%, F_6,14_ = 43.75, *p* < 0.0001, Appendix A), GST (inhibition 25.63%, F_6,14_ = 41.55, *p* < 0.0001, Appendix A), and MFO (inhibition 20.42%, F_6,14_ = 258.8, *p* < 0.0001, Appendix A). However, methyl 2-(methylamino)benzoate showed poor inhibitory activity toward GST (Inhibition 6.7%, *p* = 0.9624). In a previous study, major mechanisms of pyrethroid resistance in insects involve mutation within the target site of the insecticide and/or an increase in the activity of insecticide detoxification enzymes [39]. It is likely that the metabolic activity of detoxification enzymes was enhanced in pyrethroid-resistant mosquitoes compared with that of pyrethroid-susceptible mosquitoes [43]. We came to the same conclusion through enzymatic activity experiments (Appendix A). Therefore, CRB EO and its components can inhibit the activity of detoxification enzymes, which suggests that they can influence mosquito resistance and thus may have high potential in the control of mosquitoes.

### 3.6. Internal Microbiota Community

The diversity of the internal microbiota community was significantly affected by exposure to semi-mature EO and its main components. The *α*-diversity indices (Chao1, Shannon, and Simpson) are shown in Figure 5. The results indicated that compared with blank group mosquitoes, the exposure of resistant mosquitoes to semi-mature EO and its main components and deltamethrin led to a significant alteration in the internal microbiota diversity and evenness. An analysis using ANOVA and Tukey’s test revealed a significant difference in the Shannon (F_5,12_ = 23.84, *p* < 0.0001, Figure 5B, Appendix A) and Simpson (F_5,12_ = 18.77, *p* < 0.0001, Figure 5C, Appendix A) indices between the control group and the treated groups. The Chao1 (F_5,12_ = 197.0, *p* < 0.0001, Figure 5A, Appendix A) indices showed similar changes, except in the *p*-cymene group (df = 12, *p* = 0.5718). The results indicated that exposure of mosquitoes to different samples had an obvious effect on the biodiversity and abundance of the internal microbiota. As shown in Figure 5B, no significant difference in the Shannon indices was observed between semi-mature EO and deltamethrin (df = 12, *p* = 0.2299), D-limonene and deltamethrin (df = 12, *p* = 0.9995), and *γ*-terpinene and deltamethrin (df = 12, *p* = 0.9963). There was also no significant difference in the Simpson indices between the samples (that were made from semi-mature EO and its main components) and deltamethrin (df = 12, *p* < 0.0001).

We investigated the effect of CRB EO and its main components on the composition and relative abundance of the internal microorganisms at the phylum and genus levels. The results are shown in Figure 6. In adult mosquitoes, the majority of microorganisms are Gram-negative aerobes or facultative anaerobes, with the dominant bacteria from *Proteobacteria*, *Bacteroidetes*, *Firmicutes*, and *Actinobacteriota* at the phylum level. This result aligns with previous reports [44]. Upon exposure to CRB EO and its main components, the relative abundance of microorganisms exhibited drastic alterations compared with the blank group. In particular, the abundance of *Proteobacteria* decreased dramatically from 90% to 50% (Figure 6A).

Notably, the exposure of mosquitoes to CRB EO and its main components led to a significant alteration in the relative abundance of several core genus of the microbiota. For example, the blank group showed significant enrichment of endosymbiont *Wolbachia*, with relative abundance exceeding 60%, whereas the compound-treated group exhibited a decrease to less than 40% (Figure 6B,C). The results showed that CRB EO and its main chemical compounds could significantly reduce the abundance of *Wolbachia* (F_5,12_ = 184.5, *p* < 0.0001, Figure 6C, Appendix A), and D-limonene exhibited a comparable effect on *Wolbachia* with deltamethrin (df = 12, *p* = 0.0807). They had a significant effect on the community of internal microbiota in *Ae. albopictus*. The endosymbiont *Wolbachia* is commonly found in mosquitoes [45]. *Wolbachia* possesses the ability to stimulate the production of reactive oxygen species in mosquitoes, activating the immune system [46]. Interestingly, the abundance of *Enterobacter*, another important core strain, exhibited a different pattern of changes. That is, the relative abundance of *Enterobacter* increased from 3% to 28% (F_5,12_ = 36.75, *p* < 0.0001, Appendix A) after exposure to semi-mature EO and decreased upon exposure to D-limonene and deltamethrin (Figure 6D). This may be a consequence of the other components in semi-mature EO that have a stronger influence on the abundance of *Enterobacter*. High levels of *Enterobacter* can produce hemolytic enzymes, which can promote the digestion of food after blood sucking and play a key role in the growth, development, and life cycle of mosquitoes [47]. The relative abundance of the other two core strains, *Elizabethkingia* (F_5,12_ = 25.56, *p* < 0.0001, Appendix A) and *Pseudomonas* (F_5,12_ = 224.5, *p* < 0.0001, Appendix A), upon exposure to D-limonene, also shows an increasing trend (Figure 6E,F). This suggests that the two strains may have similar mechanisms of action that affect host life. In *Pseudomonas*, the relative abundance increased from 0.28% to 9.40% after exposure to D-limonene (df = 12, *p* < 0.0001). Zhang et al. [48] found that *Pseudomonas* was abundant in deltamethrin-susceptive strains of mosquitoes, suggesting that *Pseudomonas* is related to deltamethrin resistance. *Pseudomonas* plays a role in the metabolism of organophosphates and pyrethroids [48].

This study focused on the effect of CRB EO and its main chemical compounds on the internal microbiota community in virgin adult *Ae. albopictus*. Notably, the genus *Wolbachia* is thought to be essential for mosquitoes and shows a significant decrease after treatment with CRB EO and its main chemical compounds. Further research is required to ascertain the specific role of the altered abundance of the various bacteria discovered in this study with the aim of elucidating the mechanism underlying the microbial influence on anti-mosquito measures.

Literature reports have demonstrated that the microbiota present in insects plays a role in mediating the resistance of hosts to insecticides, and the microbiota in mosquitoes might contribute to the degradation of insecticides [49,50]. As shown in Figure 7, it is speculated that CRB EO inhibits the activity of detoxifying enzymes directly and significantly affects the microbial homeostasis to control mosquitoes. In addition, CRB EO is able to affect microorganisms within mosquitoes, which may contribute to pesticide detoxification. In such a sense, CRB EO may be utilized as a potential synergistic agent in adult mosquitoes.

## 4. Conclusions

In this study, we extracted EO from the peels of CRB at different stages of maturity and assessed the larvicidal and enzymatic activity and internal microbiota of pyrethroid-resistant *Ae. albopictus*. The results indicate that CRB EO may be used to effectively control the larvae of resistant *Ae. albopictus* and the maturity of the Chachi fruit has little effect on the toxicity of CRB EO toward the mosquito larva. The main compounds of CRB EO were able to inhibit the activity of AChE and the three detoxifying enzymes. Exposure to CRB EO led to a dramatic decrease in the relative abundance of *Wolbachia*. Enzymatic and microorganism assays suggested that CRB EO may act synergistically by inhibiting the activity of metabolic detoxification enzymes in mosquitoes and influencing the abundance of internal microorganisms, thereby reducing the detoxification of pesticides. The present findings may provide a basis for the potential use of CRB EO as a new botanical insecticide and synergist, particularly for the management of drug-resistant mosquitoes.

## Figures and Tables

**Figure 1 insects-15-00345-f001:**
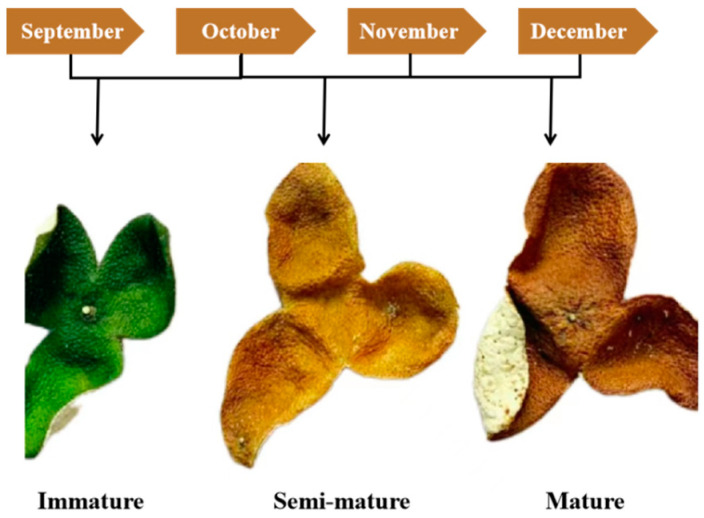
*Citrus reticulata* Blanco cv. Chachiensis (Chachi) at three ripening stages.

**Figure 2 insects-15-00345-f002:**
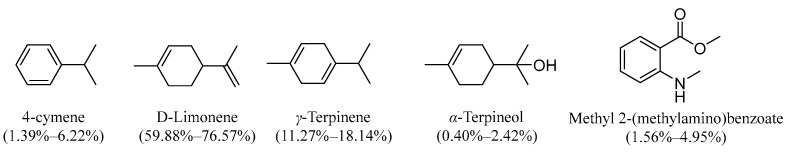
Structures of five major components in CRB EO.

**Figure 3 insects-15-00345-f003:**
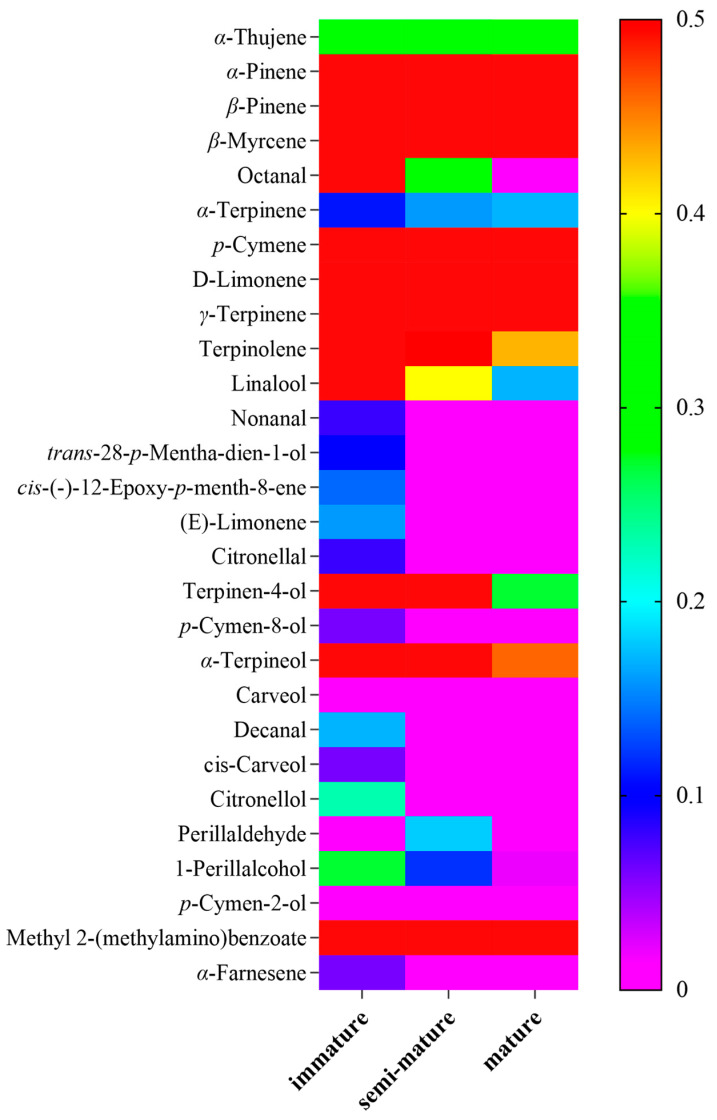
Heat map of the relative abundance levels of the components in the CRB EO.

**Figure 4 insects-15-00345-f004:**
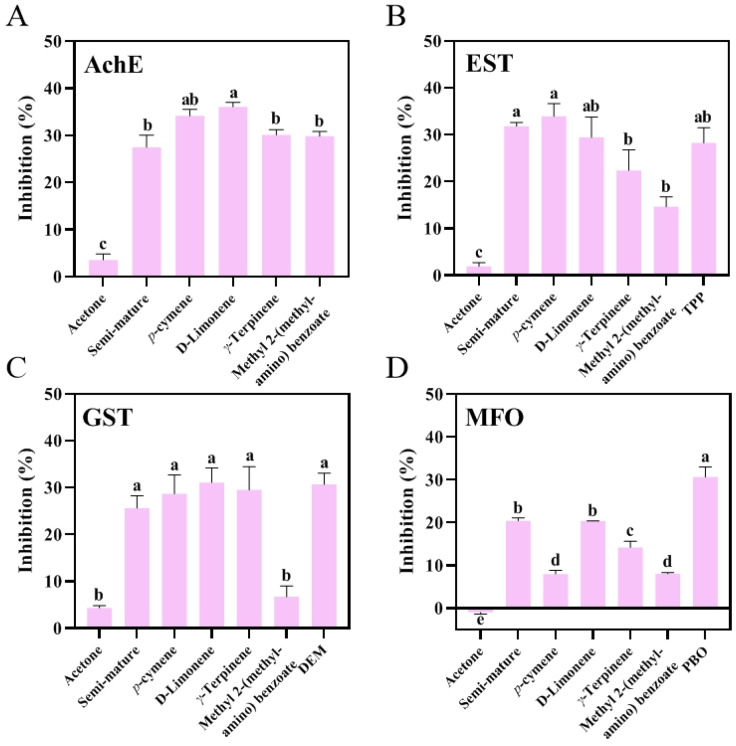
Enzyme assay in *Ae. albopictus* larvae of the resistant strain treated with CRB EO and its main chemical components. The data are presented as mean ± SD (*n* = 3). Different letters in the columns indicate significant differences between treatments. AChE, acetylcholinesterase; EST, carboxylesterase; GST, glutathione-S-transferase; MFO, multifunctional oxidase.

**Figure 5 insects-15-00345-f005:**
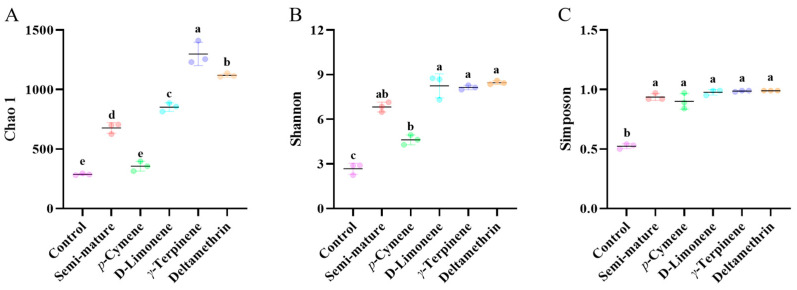
Statistical analysis of *α*-diversity indices Chao1 (**A**), Shannon (**B**), and Simpson (**C**). The abscissa represents the exposure to different samples, and the ordinate represents the value of the diversity index. The data are presented as mean ± SD (*n* = 3). Different letters indicate significant differences between treatments.

**Figure 6 insects-15-00345-f006:**
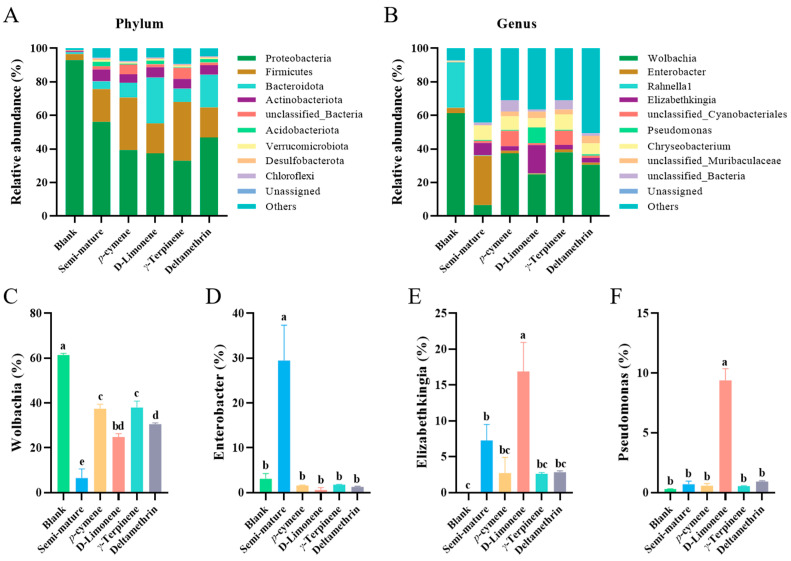
Composition and relative abundance of internal microorganisms at the levels of phylum (**A**) and genus (**B**) in each sample. The abscissa represents the different compound groups, and the ordinate represents the relative abundance of *Wolbachia* (**C**), *Enterobacter* (**D**), *Elizabethkingia* (**E**), and *Pseudomonas* (**F**). The data are presented as mean ± SD (*n* = 3). Different letters in the columns indicate significant differences between treatments.

**Figure 7 insects-15-00345-f007:**
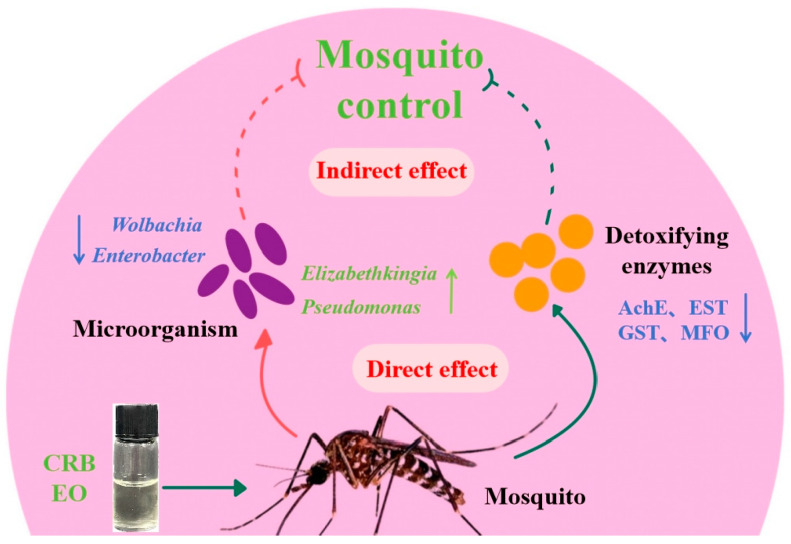
Influence of microorganisms on the detoxification and metabolism of hosts.

**Table 1 insects-15-00345-t001:** Chemical compositions of CRB EO.

No.	Component	RI		Content (%) ^3^		
		Exp. ^1^	Lit. ^2^	Immature	Semi-Mature	Mature
1	*α*-Thujene	923	924	0.29 ± 0.01	0.31 ± 0.01	0.34 ± 0.02
2	*α*-Pinene	929	929	0.87 ± 0.03	1.05 ± 0.00	0.96 ± 0.00
3	*β*-Pinene	971	971	0.92 ± 0.00	0.92 ± 0.02	0.85 ± 0.00
4	*β*-Myrcene	990	990	1.18 ± 0.02	1.19 ± 0.01	1.27 ± 0.02
5	Octanal	1001	1001	0.64 ± 0.02	0.28 ± 0.00	-
6	*α*-Terpinene	1013	1014	0.11 ± 0.02	0.16 ± 0.03	0.17 ± 0.03
7	*p*-Cymene	1022	1024	5.35 ± 0.03	2.60 ± 0.05	1.39 ± 0.00
8	D-Limonene	1030	1030	62.35 ± 1.10	76.72 ± 0.79	73.15 ± 0.09
9	*γ*-Terpinene	1062	1062	14.26 ± 0.35	11.04 ± 0.65	11.27 ± 0.12
10	Terpinolene	1086	1086	0.65 ± 0.03	0.50 ± 0.05	0.43 ± 0.00
11	Linalool	1099	1099	0.91 ± 0.02	0.40 ± 0.02	0.17 ± 0.01
12	Nonanal	1103	1102	0.08 ± 0.02	0.01 ± 0.01	-
13	trans-2,8-p-Mentha -dien-1-ol	1119	1120	0.10 ± 0.01	-	-
14	*cis*-(-)-1,2-Epoxy -p-menth-8-ene	1131	1136	0.14 ± 0.01	-	-
15	(*E*)-Limonene oxide	1136	1137	0.16 ± 0.01	-	-
16	Citronellal	1152	1151	0.08 ± 0.02	-	-
17	Terpinen-4-ol	1175	1175	1.15 ± 0.03	0.61 ± 0.01	0.27 ± 0.03
18	*p*-Cymen-8-ol	1183	1184	0.06 ± 0.02	-	-
19	*α*-Terpineol	1188	1188	2.42 ± 0.09	1.26 ± 0.02	0.46 ± 0.04
20	Carveol	1198	1200	-	-	0.01 ± 0.01
21	Decanal	1203	1203	0.17 ± 0.01	-	-
22	*cis*-Carveol	1217	1216	0.06 ± 0.00	-	-
23	Citronellol	1232	1232	0.23 ± 0.01	-	-
24	Perillaldehyde	1271	1271	-	0.18 ± 0.02	-
25	1-Perillalcohol	1297	1297	0.27 ± 0.02	0.12 ± 0.02	0.02 ± 0.00
26	*p*-Cymen-2-ol	1301	1300	-	0.01 ± 0.00	-
27	Methyl 2-(methylamino) benzoate	1405	1402	4.95 ± 0.05	1.93 ± 0.01	2.15 ± 0.20
28	*α*-Farnesene	1507	1507	0.06 ± 0.02	-	-

^1^ Linear retention index on the TG-5 MS column, experimentally determined using homologous series of C_8_–C_40_ alkanes. ^2^ Linear retention index was taken from NIST. ^3^ The EO was obtained from immature, semi-mature, and mature CRB.

**Table 2 insects-15-00345-t002:** LC_50_ values and resistance ratios associated with deltamethrin.

Ae. albopictus	Strain	LC_50_ (95% CI)/mg/L	RR	Resistance Level
Larvae	Lab-S	0.011 (0.01–0.02)	-	-
	Pyr-R	0.085 (0.05–0.13)	7.73	Low resistance
Adult	Lab-S	0.49 (0.45–0.58)	-	-
	Pyr-R	6.62 (4.11–9.30)	13.51	Mid resistance

95% CI = 95% confidence intervals.

**Table 3 insects-15-00345-t003:** Biological activity of CRB EO and five main components against pyrethroid-resistant *Ae. albopictus* larvae.

**Sample**	**LC_50_ (95% CI)/mg/L**	**LC_90_ (95% CI)/mg/L**	**χ2**	***p*-Value**
Immature	65.32 (61.63–68.89)	92.29 (86.54–100.19)	1.60	0.90
Semi-mature	61.47 (58.82–64.08)	77.40 (73.97–81.96)	6.05	0.20
Mature	65.91 (58.55–72.78)	100.21 (89.30–119.32)	8.50	0.13 n.s.
*p*-cymene	72.31 (67.08–77.39)	90.29 (83.52–102.96)	6.92	0.14 n.s.
D-Limonene	64.33 (54.75–71.90)	104.67 (92.94–126.14)	13.55	0.04 n.s.
*γ*-Terpinene	75.78 (70.25–80.89)	100.54 (93.27–111.99)	10.75	0.10 n.s.
*α*-Terpineol	>155	-	-	-
Methyl 2-(methyl- amino)benzoate	109.75 (105.98–113.58)	146.53 (139.43–156.22)	6.05	0.42

n.s. = not significant (*p* < 0.15). 95% CI = 95% confidence intervals. χ2 = chi-square.

**Table 4 insects-15-00345-t004:** Biological activity of CRB EO against pyrethroid-resistant *Ae. albopictus* adults.

Sample	LC_50_ (95% CI)/mg/L	LC_90_ (95% CI)/mg/L	χ2	*p*-Value
Immature	>2500	-	-	-
Semi-mature	2715.03 (2448.99–2991.23)	3850.04 (3412.25–4825.69)	11.20	0.02 n.s.
Mature	>2500	-	-	-
Deltamethrin	6.62 (4.11–9.30)	18.14 (11.98–65.9)	19.14	0.01 n.s.

n.s. = not significant (*p* < 0.15). 95% CI = 95% confidence intervals. χ2 = chi-square.

## Data Availability

Data are contained within the article or the Appendix A. The original contributions presented in the study are included in the article/Appendix A, and further inquiries can be directed to the corresponding author/s.

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
