# Peer review of "Chemical Composition of Essential Oil from Citrus reticulata Blanco cv. Chachiensis (Chachi) and Its Anti-Mosquito Activity against Pyrethroid-Resistant Aedes albopictus"

_insects, 2024, doi:10.3390/insects15050345_

Round 1
Reviewer 1 Report (Previous Reviewer 1)
Comments and Suggestions for Authors
The authors have made improvements to the manuscript; however, it still requires revision in several areas. Particularly, although the results are interesting, the discussion is lacking and needs to be reconsidered. Additionally, I recommend a thorough English revision, as some parts of the manuscript are difficult to understand. Furthermore, re-analyzing the data of the enzyme assays and the microbial group abundance using ANOVA instead of t-tests is suggested.
In detail:
Line 13: The authors did not calculate any diversity index for the microbiota, so they cannot assess that the biodiversity of the microbiota was reduced by the C. reticulata EOS. Also, the effect on the microbiota was assessed only for the semi-mature one, not for “Three essential oils”. Please delete or rephrase.
Lines 17-18: This study does not focus on the extraction of the EOs but rather on their toxicity against the mosquito. Please delete or rephrase.
Lines 50-60: The complete name of the species is required only the first time it is cited; hereafter, cite it as C. reticulata.
Line 51: “Pericarpium Citri reticulatae”. Please do not use italics and be consistent with acronyms and definitions. Please change “(Chenpi in Chinese)” to (PRC, Guang Chen Pi in Chinese).
Line 58: Please change “Pericarpium Citri Reticulatae (PCR, Guang Chen Pi in Chinese)” to “PCR”.
Line 121: Change “pyrethroids” to “deltamethrin”.
Lines 200-202: There are no references or data provided. Please include references, report data, or delete the sentence.
Line 204: “Significant amounts” is too vague. Is it referring to “high amounts”?
Lines 243—245: “Volatile substances” is too vague. Consider changing the sentence to: “The results indicate that those chemicals are highly lethal to Ae. albopictus larvae. Therefore, they are likely responsible for the larvicidal activity observed in the CRBs EOs.”
Line 255-256: The trial is about acute toxicity, not survival. Please change the sentence to "EOs showed low toxicity against adult mosquitoes."
Line 257-260: There is a misleading confusion between PCR and CRBs EOs. This study is about CRB EO. Please rephrase the sentence.
Lines 275-276: P-cymene and D-limonene are not essential oils. Please rephrase the sentence.
Line 277-278: “Studies on the three other detoxification enzymes showed that the activity toward EST and GST was significantly reduced. Unclear, please rephrase the sentence.
Line 279- 281: “In a previous study, major mechanisms of pyrethroid resistance in insects involve mutation within the target site of the insecticide and/or an increase in the rate of insecticide detoxification [41].” Unclear. Please check.
Lines 281-282: “Therefore, pyrethroid-resistant mosquitoes are likely to exhibit enhanced activity against metabolic detoxification enzymes, compared with pyrethroid-susceptible mosquitoes.” Unclear. How can mosquitoes exhibit “enhanced activity against metabolic detoxification enzymes”? Anyway, this experiment did not compare pyrethroid-resistant with pyrethroid-susceptible mosquitoes.
Lines 288-291: I cannot see the point. This study is not a comparison between “pyrethroid-exposed and non-exposed larvae and adults. The observed difference is between EO exposed and non-exposed mosquitoes. My suggestion is to delete the two sentences starting with “We investigated…”
Lines 303-305: This sentence is nonsensical. If I understand the statistics correctly, the authors tested the differences between the control group and each of the treatments but not among treatments. To do that, they should have analyzed data by ANOVA instead of the t-test. If they mean to say that the effects of p-cymene and D-limonene on Wolbachia are equally different from those of deltamethrin, then the t-test is not the right test to apply, and in any case, according to Fig. 5C, that is not true (blank vs p-cymene and D-limonene: p < 0.0001; blank vs deltamethrin: p < 0.001).
Line 305-306: In fact, according to Fig. 5C, not only p-cymene and D-limonene had significant effects on Wolbachia, but also the other treatments. Please reformulate.
Line 311: Not true. According to Fig. 5D, only the treatment by D-limonene induced a significant reduction in the relative abundance of Enterobacter.
Lines 315-322: Not true. According to Figures 5E and 5F, the relative abundance of Elizabethkingia and Pseudomonas did not change “in a similar fashion”. The p-cymene treatment is not significantly different from the control for the relative abundance of Elizabethkingia, while the Pseudomonas one is significantly different. On the contrary, the relative abundance of Elizabethkingia in the CRB EO was significantly different from the control but not the one of Pseudomonas. The authors should reconsider their results and better discuss them.
Line 323: Please change “insecticides and compounds” to “CRB EO and its main chemical compounds”.
Line 325: Please change “pesticide” to “CRB EO and its main chemical compounds”.
Lines 339- 341: Too speculative. This experiment did not assess any regulation of gene expression. Please consider changing the sentence to “In addition, CRBs EOs affecting microorganisms within mosquitoes that may contribute to pesticides detoxification could be utilized as potential synergistic agents for insecticidal purposes in adult mosquitoes.”
Figure 6: Consistently, delete the step of the indirect effect on “Detoxifying gene”.
Lines 348-350: Please consider changing the sentence to: “The results indicate that the essential oils from Chachi peels may effectively be used to control the larvae of resistant Ae. albopictus, and that the maturity of the Chachi fruits has little effect on the toxicity of the peels EO on the mosquito larva.
Line 350-351: Too speculative. Consider changing the sentence to: “ These results may be useful for mosquito resistance management.”
Comments on the Quality of English LanguageI recommend a thorough English revision, as some parts of the manuscript are difficult to understand
Author Response
Please see the attachment.

Reviewer 2 Report (Previous Reviewer 3)
Comments and Suggestions for Authors
Use always terpinen-4-ol (not terpin-4-ol: sorry my mistake!!!)
In Figure 3 you used 2 different nomenclature for esters: e.g. hexadecanoic acid methyl ester or ethyl linoleate: change to only 1 type of nomenclature!
Round 2
Reviewer 1 Report (Previous Reviewer 1)
Comments and Suggestions for Authors
The ms is improved but there is still the use of some very confusing terminology. In addition the anova results are not properly reported.
In particular:
1. The Authors should make an additional effort to clarify the terminology for the essential oil tested. They used the term “three essential oils” from the peels of C. reticulata. Actually, it is just one essential oil extracted from fruit at different maturity and the second part of the study is about semi-mature EO only since no significant difference was found among the three stage of maturity. For these reasons speaking of “three essential oils” may be misleading for the reader. So, for clarity, it would be better to refer to it as the essential oil of C. reticulata at three maturity stages.
2. The Authors should clearly declare in detail the aims of the study in the final period of the Introduction
3. The results of the ANOVAs are not properly reported.
In detail:
Title: Consider to change in: ” Chemical Composition of Essential Oil from Citrus reticulata Blanco cv. Chachiensis (Chachi) and its Anti-mosquito Activity against Pyrethroid-resistant Aedes albopictus”
Simple summary: I suggest to change the simple summary according to above in “The essential oil of Citrus reticulata Blanco cv. Chachiensis (Chachi) exhibited significant larvicidal activity. The EO from semi-mature fruits and its main components were able to control mosquitoes, by effectively inhibiting acetylcholinesterase and three detoxification enzymes and significantly altering the diversity of the internal microbiota in mosquitoes. “
Abstract:
Line 24. Change: “These Chachi peel essential oils” in: “Chachi essential oil”
Line 26. Change: “for the CRBs EOs from immature, semi-mature, and mature stages, respectively.” In “for immature, semi-mature, and mature CRB EO, respectively.”
Line 27. Change: “Citrus reticulata Blanco essential oils were able” in “CRB EO was able”
Introduction
Line 63-64. Please reformulate this part clearly declaring the aims of the study. “The aims of the study was…”
Results and discussion
Line 202. “Callistemon species”??
Line 231. “from other PCRs have” In this study the PCR EOs were not tested. This is an example of the confusion of the terminology above stated. Please revise.
Line 232. Please, change “the essential oil extracted from the peel of Citrus reticulata Blanco cv. Chachiensis (Chachi)” in “ CRB EO”. Once defined an acronym, please refer to it consistently along the text.
Line 234. PCR EOs were toxic
Line 274-287. The results of the anova are not properly reported both in the text and in figures 4 and 5. The anova test the oveall difference among the treatment. Then the significant difference between each single treatment and the others should be tested by a post hoc test. The Authors should report in the text the complete output of the anova (F, df, p). In the figures the asterisks are not appropriate since the P of anova indicate the differences among all treatments and not only between control and treatments (I suggest to use instead letters to report the result of the post-hoc test). Please revise.
Line 287-302. Same as above. The authors should report in the text the complete output of the anova. Besides, in Figure S1 there is not the results of the anova.
Line 303. Change: “We investigated the effect of semi-mature essential oil” in “We investigated the effect of CRB EO”. Please correct all the following accordingly.
Line 312-335. In this paragraph also the results of the statistic is not properly reported. If you have performed the anova please report the anova results. Please, revise.
Line 337. CRB EO
Line 340. CRB EO. Please correct all following accordingly
Figure 5. ANOVA post-hoc test results are not properly reported. See above.
Figure 6 ANOVA post-hoc test results are not properly reported. See above.
Conclusions
Line 364. Please consider to change the phrase in “In this study, we extracted the essential oil from the peels of Citrus reticulata Blanco cv. Chachiensis (Chachi) at different stages of maturity, and assessed its larvicidal, enzymatic activity and effects on the internal microbiota toward pyrethroid-resistant Ae. albopictus.”
Comments on the Quality of English LanguageEnglish is improved. Only moderate editing is required
Round 3
Reviewer 1 Report (Previous Reviewer 1)
Comments and Suggestions for Authors
The Authors have revised the ms as suggested. It is now, in opinion, suitable for pubblication
Author Response
Thank you very much for your strong recommendation. We are greatly grateful for your numerous constructive comments that have helped to improve the quality of our paper.
This manuscript is a resubmission of an earlier submission. The following is a list of the peer review reports and author responses from that submission.
Round 1
Reviewer 1 Report
Comments and Suggestions for Authors
The manuscript describes several experiments aimed at investigating the toxicity of essential oils extracted from the peels of Citrus reticulata Blanco cv. Chachiensis at three stages of maturity (immature, semi-mature, and mature), and four of their main components against larvae and adults of a Pyrethroid-resistant strain of the mosquito Aedes albopictus. The authors also evaluated the inhibition of acetylcholinesterase and three other detoxification enzymes by semi-mature citrus essential oil (EO), as well as the change in the diversity of the internal microbiota in mosquitoes treated with the EO.
While the manuscript presents some interesting results, it lacks a well-defined aim. The rationale for testing the citrus EO at three stages of maturity is weak, and the description of the experiments is missing important information.
Specifically:
The authors should clearly declare the aims of the study in the introduction.
The chemical composition of the three essential oils, especially for the main components, is very similar. So why to test all three of them? No surprise that the toxicity are not significantly different. Consider justifying the need for testing all three.
In the description of the "Internal microbiota community" experiment, it is not clear if a method was applied to separate the "internal" from the "external" microbial community. Without this clarification, the results may be unreliable. Additionally, the description of the "metagenomic analyses" is absent, and there is no mention of the methods for subsequent data analyses, which is unacceptable.
Conclusions are, in my opinion not fully supported by the results.
Other minor points:
Figure 3 should be deleted as the data are already reported in Table 1.
Table 3 is missing the p-value of the fitting of the Probit model to data (Pearson goodness-of-fit test).
According to Table 3, D-limonene and 4-cymene are not significantly different from γ-terpinene (95% CI overlapping).
The statement "However, effective essential oils against adult mosquitoes have been rarely reported" is not accurate.
The asterisks in Figures 4 and 5 are not consistent with the values, and the results of the statistical analyses are not reported in the results section.
Reviewer 2 Report
Comments and Suggestions for Authors
The manuscript is interesting and well-written. I have only a few relatively minor corrections/suggestions.
title line 4 consider changing for to against
Line 15 change drug to insecticide
line 22 omit "the"
Table 1 are the contents %s means? if so, how many reps?
Table 3 what is the meaning of the Chi-square values?
line 309 the eradicate seems a little strong, consider replacing with better term
Reviewer 3 Report
Comments and Suggestions for Authors
Good and interesting manuscript with minor corrections, only:
Use the correct IUPAC-names of identified compounds as follows:
4-cymene is p-cymene, 4-terpineol is terpin-4-ol, 1-methyl-4-(1-methylethenyl)-12-cyclohexadienediol is 12-cylohexandediol-methyl-4-(1methylethenyl), (Z)-9-octadecenoic acid methyl ester is 9-octadecenoic acid-(Z) methyl ester.
2 some minor mistakes were found and should be added or changed, as follows:
Page 3, 2.1. Plant material and chemicals: add voucher speciman for botanical identification.
Page 6, Table 2: change Larave to Larvea.
Reviewer 4 Report
Comments and Suggestions for Authors
The paper investigates the biocidal activity of essential oils extracted from peel fruits of Citrus reticulata Blanco cv. Chachiensis (Chachi) at various maturity stages. The study is well-executed, and the data are clearly presented. However, the introduction could be more elaborate in emphasizing the objectives of the work. Overall, the manuscript would benefit from the input of a native speaker after revision.
Some specific comments:
Line 20-21 what are these range of percentages indicating? Maybe the range across the immature and mature, but it should be rewritten better giving the right details.
Line 23, again what are these range indicating?
Line 23, Citrus reticulata in italic.
Line 43, why insecticidal and larvicidal? Are larvae not insects themselves?
Line 49, check italic and capital letters
Line 51,53, reticulate or reticulata? Here and across the manuscript
Line 89, depending on what?
Line 112-114, there is no need to report how LRI were calculated.
Line 135, correct capital letter
Line 150, check
Line 162, this is more a “Results and discussion” section rather than a “results” section.
Table 1, why is LRI of D-limonene missing? In literature is present.
Figure 2 is unnecessary
In table 1 the values of standard errors of the means are missing. For the chemicals analysis how many replications were carried out?
Line 214,220, check capital letters, here and across the paper.
Line 223, not totally true so the author should dampen this sentence; give some references, for example:
Lucia, A., Licastro, S., Zerba, E., Audino, P. G., & Masuh, H. (2009). Sensitivity of Aedes aegypti adults (Diptera: Culicidae) to the vapors of Eucalyptus essential oils. Bioresource technology, 100(23), 6083-6087.
McAllister, J. C., & Adams, M. F. (2014). Mode of action for natural products isolated from essential oils of two trees is different from available mosquito adulticides. Journal of medical entomology, 47(6), 1123-1126.
Line 308-309, again check for capital letters unappropriated.
Line 316-318, rewrite this sentence, here was not tested as pesticide synergist.
Comments on the Quality of English Languageaverage English style, could benefit of a review from amother tongue entonomologist.
